# Cyber Dating Abuse and Masculine Gender Norms in a Sample of Male Adults

**Beatriz Villora** * , **Santiago Yubero and Raúl Navarro** *

Deparment of Psychology, Faculty of Education and Humanities, University of Castilla-La Mancha,
16071 Cuenca, Spain; santiago.yubero@uclm.es
* Correspondence: beatriz.villora@uclm.es (B.V.); raul.navarro@uclm.es (R.N.)

**Abstract:** Gender role norms have been widely studied in the offline partner violence context. Different studies have indicated that internalizing these norms was associated with dating violence. However, very few research works have analyzed this relation in forms of aggression against partners and former partners using information and communication technologies (ICT). The objective of the present study was to examine the co-occurrence of cyber dating abuse by analyzing the extent to which victimization and perpetration overlap, and by analyzing the differences according to conformity to the masculine gender norms between men who are perpetrators or victims of cyber dating abuse. The participants were 614 male university students, and 26.5% of the sample reported having been a victim and perpetrator of cyber dating abuse. Nonetheless, the regression analyses did not reveal any statistically significant association between conformity to masculine gender norms and practicing either perpetration or victimization by cyber dating abuse.

**Keywords:** cyber dating abuse; masculine gender conformity; gender role attitudes; university students

---

## 1. Introduction

The use of information and communication technologies (ICT) has exponentially grown in recent years. In 2017, 66% of the people worldwide owned a mobile phone and Spain led the world ranking with 88% of users. Nowadays, a mobile phone is the device most widely used to access the Internet, with a mean frequency of use of 170 min a day [1]. In fact, using the Internet has become a habit practiced by the majority of youths aged between 16 and 24 years, with minor differences between men (98.3%) and women (98.7%) [2]. The study performed by Hootsuite [3] indicates that 92% of the Internet users in Spain connect on a daily basis and spend a mean of 5 h and 20 min browsing the Internet from any device, and especially spend time using social networks, for which they spend a mean time of 1 h and 50 min.

In line with former research, ICT like smartphones (text messages, messenger services), websites of social networks (Facebook, Twitter), and telephone applications (Instagram, Snapchat), facilitate constant communication between partners [4], but can also be used as a means to directly exercise aggression and to control partners or former partners [5]. Although various terms can be used to name this phenomenon, such as "electronic dating violence" [6], "digital forms of dating violence" [7], "intimate partner cyber harassment" [8], "cyber partner abuse" [9], "cyber psychological abuse" [10], the term most widely used by the scientific community is "cyber dating abuse" [11,12].

Cyber dating abuse includes direct aggression conducts that intend to deliberately harm one's partner (e.g., sharing private information and posting it on technological platforms, and insulting or threatening using these platforms), but also includes control behaviors that invade people's privacy or are related to monitoring someone (e.g., controlling access to social networks or using one's partner's

password without his/her permission) [9]. These strategies often work as a power and control mechanism within relations, and are frequently motivated by jealousy and mistrust [13].

Although cyber dating abuse forms part of a broader violence connection and is related to the partner's personal experiences of psychological, physical, and sexual aggression [14], both behaviors differ in several aspects. Regarding the differences between one aggressive behavior and another, research has shown that in digital contexts (online) 1) less blame is experienced by the perpetrator [15]; 2) retaliation may be more usual than in offline contexts as some victims can be more able to counter-attack thanks to the security of being protected by a screen [16]; 3) lack of physical and time limits means that aggression can be shown more easily [8,12]; 4) similarly, the constant interaction on the Internet makes monitoring victims easier [13]; and 5) the difficulty of observing the victim's emotional response could reduce perpetrators' capacity to accurately assess the harm that their actions cause which, therefore, diminishes their inhibition [17]. These differences make further research into digital behaviors necessary.

### 1.1. Risk Factors for Engaging in Dating Abuse

Although offline aggression risk factors have been widely studied, studies about the risk factors in perpetration and victimization by online aggression are relatively scarce [18]. Among the variables most well-studied, many studies have examined the influence of being previously involved as victims or perpetrators as being a risk factor to exercise perpetration or to suffer victimization by offline abuse in partner relationships [19–21]. Longitudinal research has demonstrated that a reciprocity relation exists between both roles [22], and this relation takes place regardless of the participants' gender. In other words, being a victim of offline abuse in partner relationships increases the probabilities of becoming a perpetrator, and vice versa. Similarly, previous research that has analyzed online abuse has also found a strong correlation between perpetration and victimization of cybernetic aggression [4], which suggests that online aggression also tends to be overlapping in nature [11,23,24].

Many other studies have analyzed the differences in implication depending on the participants' gender, but their results have provided mixed results. Some researchers indicate that men have higher perpetration rates [25], while others have found higher victimization rates for men [26]. Other studies contradict these results and state that women present higher perpetration [18,27] and victimization [27–29] rates, or have found no significant differences between genders [23]. If we contemplate these contradictory results, it is important to go beyond merely analyzing gender differences and analyze which socio-cultural factors linked to gender socialization are related to the fact that these differences appear or not [30]. Indeed, research suggests that adhering to conventional gender roles can more strongly impact aggressive behaviors than biological sex [31].

### 1.2. Gender Norms and Dating Abuse

One risk factor that is frequently theorized for men's offline aggression toward their partners is sticking to conventional gender roles. The gender role theory indicates that "gender roles set socially constructed expectations and norms about appropriate male and female behavior, characteristics, roles, and the culturally acceptable dynamics between males and females" [32]. Indeed, the review by Stith et al. [33] revealed that one of the risk factors in men for exercising physical perpetration to an intimate partner is the traditional masculine gender role ideology, although with a moderate effect size (r = 0.29).

Gender role norms can be defined as a set of rules that guide and prescribe what men must do, think or feel [34]. Conformity to gender role refers to the extent to which men agree with or meet the gender expectations that their culture imposes [35]. From this point of view, gender is inherently linked to contextual influences [36] and the degree to which gender roles are emphasized in a given context depends on the culture examined [37]. In a cross-cultural study involving 25 countries, Williams and Best [38] found variations in the internalization of gender roles that were attributable to cultural differences. However, the major finding was a high degree of pancultural similarities in the gender

roles differentially associated with women and men in the 25 countries studied. Cuéllar-Flores et al. [39] found that male university students in Spain reported lower conformity rates of gender norms than North American university students in dimensions such as disdain for homosexuality, the importance of winning, pursuit of status, attraction of violence, power over women desires, and risk-taking behaviors. Nevertheless, students in Spain reported greater conformity regarding searching for multiple sexual relationships without commitment.

Masculine gender norms are related to social command, power over women, violence, and self-sufficiency [34]. If power and control are the factors stressed in the construction of masculinity, female-related abusive or controlling behaviors can be congruent with this construction [40]. Indeed, traditional masculine gender roles have been related to aggression between partners [41], which suggests that men exercising abuse have extreme masculine identification and inflict abuse to show their masculinity. Along the same lines, previous studies confirmed that supporting more traditional masculine gender roles on men is a predictor of the perpetration of psychological violence [42].

Conversely, the men who present low levels of conformity to these norms are more prone to behave prosocially and to display less aggression [43]. However, other studies have found that men on the opposite end of the conformity continuum with gender roles (men who adhere less to masculine norms), can also participate in aggressive and violent behaviors in romantic relationships as they do not conform to the ideal manhood that stems from social norms [32]. In such cases, aggression can be a mechanism to overcome their lack of adapting to traditional gender norms or to defend themselves from attacks to their masculinity [44].

Along these lines, Eisler et al. [45] found that men who did not conform to traditional masculine norms experience gender role stress and more negative reactions in their intimate relationships (e.g., irritation, jealousy, and rage). Other studies have found that gender role stress was related to control behaviors in a sample of male perpetrators against women [40]. However, Reidy et al. [32] found no interaction between discrepancies with gender roles (i.e., being less masculine than the typical man) and the stress that emerges from this discrepancy (i.e., anxiety from being less masculine) with perpetrating dating violence. Although they did find a relation between gender role stress and a higher risk of perpetrating sexual violence.

The relation between gender role norms and cyber dating abuse has been less analyzed than offline abuse. Regarding the perpetrator's role, in secondary education students, [46] found that males who endorsed stereotypical gender beliefs were more likely to perpetrate sexual coercion and direct aggression. Van Ouytsel, Ponnet, and Walrave [47] discovered that adolescents endorsing gender stereotypes reported significantly higher perpetration control behaviors. Martínez-Pecino and Durán [48] found that men with high hostile sexism levels reported higher cyber abuse in romantic relationships compared to those with low hostile sexism levels.

As far as we know, the relation between cyber dating abuse victimization and conformity to gender norms has not yet been explored. Nevertheless, previous studies that have analyzed the relation between conventional gender norms and bullying behaviors have reported that youths not conforming to traditional masculine gender norms were more victimized by their peers [49,50].

The present study had three objectives: (1) Present prevalence rates for cyber dating perpetration and victimization in a sample of male university students. Based on the previous studies [19,23] it was expected that male university students will report different experiences of victimization and perpetration and that the most frequent form of abuse will be the control abuse. (2) Explore cyber dating abuse's co-occurrence by analyzing to what extent victimization and perpetration may overlap. Given previous research [4,22], a correlation was expected between cyber dating victimization and perpetration in both control and direct forms of abuse. (3) Examine the differences in conformity to masculine gender norms among men who identify as perpetrators or victims of cyber dating abuse. Considering previous research, it was expected that males showing higher conformity to masculine gender norms will be more involved as perpetrators of cyber dating abuse [46,47]. On the contrary, males not conforming to traditional gender roles will report higher victimization [50].



## 2. Materials and Methods

### 2.1. Participants

Cross-sectional analyses were conducted using the data from a non-probabilistic sample collected from February to March 2018 in a Spanish university that serves a student body of approximately 23,000 students and located in central Spain. The male university student population in the 2017/2018 academic year was 9725 undergraduate students. Sample size was calculated considering a Z value of 1.96 (95% confidence level) and ±4% error margin with an expected proportion (P) of 0.5. The analyses determined that 566 students were required for the study. We intentionally oversampled and obtained data from 655 undergraduate students. After eliminating two participants because they did not provide valid data for all the study variables, the sample was made up of 663 participants. The data analysis was performed with those participants who indicated having had a romantic relationship in the last year or being presently involved in a romantic relationship. Forty-nine cases were excluded for that reason. The final sample was made up of 614 undergraduate men. Participants' ages ranged from 18 to 42 years old (*M* = 20.73; *SD* = 3.54). Of all the participants, 54.6% were studying Social Sciences, 28.7% Applied Sciences, and 16.8% Physical and Health Sciences.

### 2.2. Measurement Variables and Instruments

The participants provided information about demographic variables, such as age and sexual orientation. The following instruments were used to analyze the study variables.

*Cyber dating Abuse.* The cyber dating abuse (CDA) questionnaire [23] consists of 20 items on different CDA types. It includes different online abusive behaviors from victimization to perpetration perspectives. The questionnaire consists of two factors: direct abuse, including aggressive acts with the deliberate intention to hurt one's partner/former partner (item example: "I threatened my partner or former partner using new technologies to physically hurt her/him"), and control abuse that refers to using electronic means to control one's partner/former partner (item example: "By mobile applications, I controlled the time that my partner or former partner last connected"). Item scored on a 6-point scale: 1 (*never*); 2 (*not in the last year, but before*); 3 (*once or twice*); 4 (*3 to 10 times*); 5 (*10 to 20 times*); 6 (*more than 20 times*). In the current sample, a Cronbach's alpha of 0.84 and McDonald's omega of 0.85 was found for the perpetration scale. A Cronbach's alpha of 0.82 and McDonald's omega of 0.83 was determined for the victimization scale.

*Masculine Gender Norms.* To assess participants' conformity to masculine norms, we used the abbreviated version of the conformity to masculinity norms inventory (CMNI) [34,51], adapted to Spanish by [50]. The inventory consists of 46 items answered on a 4-point scale (0 _strongly disagree_ to 3 _ strongly agree) and assesses conformity to an array of masculinity norms found in the US society. Masculine gender norms are grouped on nine scales: 1) Emotional control; 2) winning; 3) playboy; 4) violence; 5) self-reliance; 6) risk-taking; 7) power over women; 8) primacy of work; 9) heterosexual self-presentation. Reliability in the current sample yielded a Cronbach's alpha coefficient of 0.72, and a McDonald's omega of 0.83 for the total scale. Cuéllar-Flores, et al. [39] have confirmed the adequacy of the instrument in the Spain population.

### 2.3. Procedure

Data were collected by self-reported group class-administered pencil-and-paper questionnaires. Students' willingness to start the survey was taken as their implied consent. One researcher administered questionnaires to participants, explained the meaning of certain items, and answered any questions whenever they arose. Participants were also assured that their answers would remain anonymous and they could withdraw their participation at any time. The procedure took approximately 15 min in each group class. The data collecting procedure was conducted during the established schedules in all classrooms, which took the research team 8 weeks. The study was

conducted in line with the legal requirements of the study country and in compliance with IRB approval from the (hidden for peer review).

*2.4. Analysis Plan*

We first detailed the general description of the independent variables. Second, we examined the descriptive data related to participants' involvement in cyber dating abuse (objective 1). Participants' categorization as victims, perpetrators, or perpetrators-victims in each form of abuse was done by following a criterion used by previous cyber dating researchers [23]. Participants who indicated suffering, but not perpetrating one or more times in at least three of the abusive behaviors included in the questionnaire were classified as victims. Participants who reported perpetrating, but not suffering, one or more times in at least three of the abusive behaviors were classified as perpetrators. Participants who indicated suffering and perpetrating one or more times on at least three of the abusive behaviors were classified as perpetrators-victims. The remaining students were considered not involved in cyber dating abuse. Third, to evaluate the relationship between cyber dating perpetration and victimization during the last year (objective 2), perpetration and victimization behaviors were correlated on the direct and control subscales and total score (sum of direct and control abuse). Finally, hierarchical regressions were performed to determine if the nine subscales of masculine gender norms were associated with cyber dating abuse (by analyzing direct and control forms of abuse), over and above the experience of being a victim or a perpetrator of cyber dating abuse (objective 2 and 3). Data were analyzed using SPSS Version 24.00.

## 3. Results

*3.1. General Descriptive and Prevalence Rates of Cyber Dating Abuse*

Table 1 provides the means and standard deviations of each study's variable for the whole sample. The prevalence rates for perpetrators and victims of cyber dating abuse are presented in Table 2. While half the sample reported not being a victim or perpetrator of cyber dating abuse in their relationships, 26.5% of the sample reported direct and control abuse as being overlapping (both perpetrating and receiving abuse behaviors). On average, men reported having perpetrated acts of abuse slightly less ($M = 1.24$, $SD = 0.41$) than having been the victims of abuse in the past year ($M = 1.37$, $SD = 0.48$), $t(614) = -7.00$, $p <. 001$, $d = -0.29$. Pearson correlations were conducted for further analyses. The correlation between the overall perpetration of cyber dating abuse and the overall victimization of cyber dating abuse was 0.613 ($p < 0.001$). The correlation between perpetrating direct abuse and being a victim of direct abuse was 0.553 ($p < 0.001$), and between perpetration and victimization of control abuse was 0.591 ($p < 0.001$).

**Table 1.** Summary statistics of the study variables.

| Measures | Mean | SD | Range |
|---|---|---|---|
| Age | 20.73 | 3.54 | (18–42) |
| Sexual Orientation 90.1% Heterosexual | - | - | - |
| Direct perpetration | 1.10 | 0.30 | (0–3) |
| Control perpetration | 1.37 | 0.62 | (1–5) |
| Direct victimization | 1.20 | 0.50 | (1–4) |
| Control victimization | 1.54 | 0.83 | (1–6) |
| Masculinity norms | - | - | - |
| Winning | 9.35 | 3.09 | (0–18) |
| Emotional Control | 7.29 | 2.19 | (2–14) |
| Primacy of work | 5.52 | 1.62 | (1–11) |
| Risk Taking | 8.20 | 2.16 | (0–14) |
| Violence | 6.16 | 2.82 | (0–16) |
| Heterosexual self-presentation | 5.55 | 3.35 | (0–16) |
| Playboy | 3.96 | 2.31 | (0–12) |
| Self-Reliance | 5.72 | 2.07 | (1–12) |
| Power over women | 4.39 | 1.39 | (0–8) |

**Table 2.** Prevalence rates of cyber dating abuse.

| Cyber Dating Abuse | Direction of abuse | | | |
|---|---|---|---|---|
| | None | Victimization and Perpetration | Victimization Only | Perpetration Only |
| Forms of abuse | n (%) | n (%) | n (%) | n (%) |
| Direct abuse | 499 (81.3) | 37 (6) | 57 (9.3) | 21 (3.4) |
| Control abuse | 353 (57.5) | 147 (23.9) | 75 (12.2) | 39 (6.4) |
| Total | 334 (54.4) | 163 (26.5) | 77 (12.5) | 40 (6.5) |

### 3.2. Associations between Masculinity Norms Inventory (CMNI) Scales and Cyber Dating Abuse Victimization

A hierarchical regression analysis indicated that perpetrating direct cyber dating abuse accounted for 30% of the variance in the scores related to being a victim of direct cyber dating abuse ($B = 0.895$, $SE = 0.056$, $p < 0.001$). However, after accounting for direct perpetration, the CMNI scales were not significantly associated with direct cyber dating victimization ($r^2$ change: 0.006, $p = 0.45$).

Likewise, hierarchical regression analysis indicated that perpetrating control cyber dating abuse accounted for 35% of the variance in scores related to being a victim of control cyber dating abuse ($B = 0.797$, $SE = 0.092$, $p < 0.001$). However, after accounting for control perpetration, the CMNI scales were not significantly associated with control cyber dating victimization ($r^2$ change: 0.007, $p = 0.72$).

### 3.3. Associations between Masculinity Norms Inventory (CFNI) Scales and Cyber Dating Perpetration

A hierarchical regression analysis indicated that being heterosexual and a victim of direct cyber dating abuse accounted for 31% of the variance in scores related to perpetrating direct cyber dating abuse. After accounting for direct victimization, age, and sexual orientation, only two of the CMNI scales were significantly associated with direct cyber dating perpetration, but the relationships were very weak and only accounted for an additional 1.7% of the variance in the direct cyber dating abuse (see Table 3). Higher self-reliance was related to higher direct perpetration levels, and less emotional control was related to higher direct perpetration levels.

**Table 3.** Hierarchical regression analysis analyzing the association between masculine norms and direct cyber dating perpetration.

| Variable | B | SE B | β | $R^2$ Change |
|---|---|---|---|---|
| Step I | | | | |
| Direct victimization | 0.333 | 0.021 | 0.543 *** | |
| Age | −0.001 | 0.003 | −0.012 | 0.313 |
| Sexual orientation | 0.090 | 0.035 | 0.088 ** | |
| Step II | | | | |
| Masculine norms subscales | | | | 0.017 |
| Winning | 0.005 | 0.021 | 0.539 | |
| Emotional Control | −0.011 | 0.006 | −0.081 * | |
| Primacy of work | −0.010 | 0.008 | −0.046 | |
| Risk Taking | 0.006 | 0.005 | 0.045 | |
| Violence | −0.002 | 0.004 | −0.016 | |
| Heterosexual self-presentation | 0.016 | 0.006 | 0.110 | |
| Playboy | −0.001 | 0.005 | −0.004 | |
| Self-Reliance | 0.016 | 0.006 | 0.110 ** | |
| Power over women | 0.004 | 0.007 | 0.023 | |

* $p < 0.05$; ** $p < 0.01$; *** $p < 0.001$.

A hierarchical regression analysis indicated that being a victim of control cyber dating abuse accounted for 36% of the variance in scores related to perpetrating control cyber dating abuse ($B = 0.435$, $SE = 0.024$, $p < 0.001$). However, after accounting for the control victimization received, the CNNI

scales were not significantly associated with perpetrating control cyber dating abuse ($r^2$ change: 0.005, $p = 0.57$).

## 4. Discussion

This study analyzed the association between cyber dating abuse and conformity to masculine gender norms in a sample of Spanish male university students. The objective was to extend research about cyber dating abuse by analyzing the co-occurrence of the victimization and perpetration of cyber dating abuse and by examining the differences in conformity to the masculine gender norms among those men who were perpetrators or victims of cyber dating abuse.

The results revealed that 26.5% of the participants were involved in cyber dating abuse as perpetrators/victims as opposed to 12.5% who were only victims and 6.5% who were only perpetrators. Supporting Hypothesis 1, the present study found high prevalence rates of cyber dating abuse, with a higher frequency of control behaviors. In line with previous research, the most common forms of cyber dating abuse were those that involve some form of control, such as monitoring victims' cellphone and social networks [11]. The regression analysis results indicated that being a victim of cyber dating abuse increased the probabilities of being a perpetrator, and vice versa. These results confirmed our hypothesis and are in line with previous studies showing that cyber dating abuse is a co-concurrent aggressive behavior. Our results are also in line with previous research showing that the main risk factor for suffering victimization was being formerly involved as a perpetrator, whereas the main risk for perpetration was having suffered victimization [11,23]. However, our cross-sectional study design did not allow us to confirm the causal direction of the relations found between both roles within cyber dating abuse. Violence co-occurrence may be explained by the social learning theory according to what behaviors are learned through social interactions [52]. In this sense, Gray and Foshee [53] found that youth who reported being a victim and a perpetrator also reported to being involved in a greater number of violent couples in the past. Co-occurrence could also be explained as a consequence of an escalation of violence among partners, where abuse could be a way to self-defense or retaliation [54]. Future research should analyze more carefully the factors behind the co-occurrence of cyber dating abuse.

Previous research reported a relation between internalizing gender norms/roles and different forms of perpetration and victimization in romantic relationships. However, in contrast to what we hypothesized, we neither found significant associations between conformity to masculine gender norms and victimization by cyber dating abuse in both its direct and control forms, nor associations between perpetration by control and conformity to masculine gender norms. The only significant relationships were found in direct perpetration. However, the poor statistical power of the relations found did not allow us to confirm a significant association. Notwithstanding, these results are in line with the longitudinal work by Foshee et al. [55], who found that internalizing gender stereotypes was not a predictor of perpetrating dating violence among male adolescents.

There may be several explanations for the results obtained. First of all, we must consider that participants in the current study reported low scores of conformity to masculine gender role norms, which could indicate a break from the traditional gender paradigm brought about by the socio-cultural changes that have taken place in Spain in recent times. Indeed, the study by López-Sáez, Morales, and Lisbona [56] has previously confirmed a reduction in gender role stereotypes in a representative sample of the Spanish population. Although there are still many gender inequalities, the European Union and, specifically, Spain have made important progress in order to reach higher levels of gender equality [36]. Moreover, the traditional gender socialization in terms of stereotypical feminine and masculine roles has gained flexibility during the last decades due to social, economic, and cultural changes [57]. In this sense, Cuéllar-Flores et al. [39] found that Spaniards show less conformity to traditional gender norms than North Americans except for the norm related to the search for multiple sexual relationships without commitment (playboy ideal). They suggested that these results are related to differences between cultures rather to variables such as age and educational level. However, age

and educational level are important to understand the degree of adherence to gender roles. Research has shown that gender roles and stereotypes associated with men and women begin to lose relevance as age increases, in line with moral development [39,58]. Educational level has also been related to a low endorsement of stereotypes and gender roles. For example, research has found that a higher level of education is related to lower levels of sexism [58,59].

Moreover, previous research has found that men who indicate they poorly conform to masculine norms may be more prone to behave more prosocially instead of aggressively [43]. However, many male university students in the present study reported having exercised/suffered direct or control cyber abuse in the last 12 months. These paradoxical results could be related to the fact that cyber dating abuse might not be perceived by male university students as aggression or at least as behavior that actually causes real harm. Indeed, several studies have indicated how some youths perceive these conducts as a normative part of dating instead of disruptive and harmful behaviors [11,60]. Additionally, previous studies have also shown that the influence of internalizing traditional gender roles in displaying aggressive behavior to one's partner might be moderated by other factors, such as acceptance of dating violence beliefs. For example, a longitudinal study revealed that internalizing gender stereotypes was associated only with perpetrating offline violence in romantic relationships, in those adolescents who reported high levels of internalizing beliefs about violence being standard and acceptable [61]. Future research must confirm these same relationships in digital forms of abuse.

The interpretation of the results found must consider the limitations of this study. Although our analysis of the association between conformity to masculine norms and cyber dating abuse include an analysis of the co-occurrence between victimization and perpetration, other variables need to be included [62]. Different combinations of variables would allow us to better understand the associations between gender norms and aggressive behavior. The study design is cross-sectional and, therefore, causality cannot be inferred from our findings. Moreover, perpetration and victimization of cyber dating abuse were measured by self-reporting, so the results could be biased by social desirability. Data collection was limited to a single partner of the romantic relationships, which did not allow us to know if the obtained relationships would differ if data about the reciprocity of the aggression and victimization between the partners forming the relationship is gathered [22]. Finally, as our sample was basically made up of white, heterosexual male university students from only one area in central Spain, we cannot generalize our results to other populations. Future research should include bigger samples to make intra- and inter-group comparisons among university students from ethnic minorities and sexual minorities.

Despite the limitations, our study offers new insights on the cyber dating abuse research, showing that this type of abuse is spread among young males of the Spanish university population. Our findings also suggest that males are not only perpetrators of cyber dating abuse but also victims, although the relationships between both roles and traditional gender roles in the digital environment is not as clear as in the offline environment Considering the present study results, and in line with Reidy et al. [32], prevention and interventions programs may not be effective "if they use a unilateral approach that attempts to move males from one extreme of the gender role spectrum to the other" (p. 623). Intervention models that only work to raise awareness about the negative consequences of traditional gender role socialization may be insufficient to address cyber dating abuse among university students. Prevention programs should attend to the co-occurrence nature of the abuse among young couples. In this sense, prevention efforts should be directed to both victims and perpetrators and consider the double role developed for some individuals, focusing on the impact that their own behaviors could have on their partner's behaviors [22]. Consequently, prevention programs should provide strategies to learn how to handle specific situations among partners that could trigger abusive behaviors [63].

**Author Contributions:** Conceptualization, B.V. and R.N.; data curation, B.V.; formal analysis, R.N.; funding acquisition, B.V. and S.Y.; investigation, B.V.; methodology, S.Y. and R.N.; project administration, S.Y.; Supervision, R.N.; writing—original draft, B.V. and R.N.; writing—review and editing, B.V., S.Y., and R.N.

**Funding:** This research received no external funding.

**Conflicts of Interest:** The authors declare no conflict of interest.

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
