# Peer review of "Cyber Dating Abuse and Masculine Gender Norms in a Sample of Male Adults"

_futureinternet, doi:10.3390/fi11040084_

Round 1
Reviewer 1 Report
This manuscript is an interesting contribution to the field of cyber datying abuse, especially by Gender Role Theory.
The work is well written and has my general recommendation to be accepted with major changes.
Comments
INTRODUCTION
It is well written and contains an adequate presentation of the topic of study. The main problem is in the hypothesis approach. I think it would be more appropriate to make several hypotheses (since there are several objectives) and also indicate whether these hypotheses are related to other similar studies (add intratext quotations, if so).
I am also concerned, in relation to Gender Role Theory, about the relationship that may exist with the European-Spanish context. I appreciate that several of the studies are carried out in the USA and gender theory can be closely associated with social/cultural changes. This could be further explored in discussion as well.
METHODS
Several comments:
1. Could you specify on line 133 whether it is 230,000 students? The normal thing is to indicate the number of students of the degrees where the sample has been obtained. I do not understand the data.
2. It is important to indicate the type of sampling (I imagine not probabilistic of incidental type).
3. I am concerned about age range. If there is a large sample older than 25-30 years it may be necessary to consider specific analyses, as gender theory may be affected by this variable.
4. I suggest additional reliability analyses for the sample, as there are some especially low ones (such as the CMNI). They can perform ordinal alpha or omega coefficients. Perhaps all three would give a more complete information.
5. The indicated time (40 minutes) does not make sense for the indicated questionnaires. It may be a misprint of a more ambitious project with more evaluation tools, but not for 3 questionnaires. Please review it.
6. I think it's important to further clarify the information on data analysis. I suggest making this point according to each objective and including all the analyses carried out.
It is not clear how the prevalence of the problem is calculated.
It is also not clear (and it appears in more parts) the issue of "bidirectionality". For example, the abstract reads: "bidirectionality of cyber dating abuse by analyzing the extent to which victimization and perpetration co-occur". I believe that using two profiles together is not the same as establishing bidirectionality relationships (which require other analyses).
7.On line 176, you indicate "feminine gender". This baffles me very much.
Results
8 ) I don't understand "Hetero-LGB" in table 1.
9) I did not find how the prevalence was estimated. Scores equal or above 2? Is that the usual criterion for CDA or is it provided by the original authors?
10) Why do you not use control perpetration for the victimization prediction and the same goes for direct victimization and perpetration.
11) Include the results of the cohen d.
Discussion
12) If the authors add more hypotheses they should make those adjustments in the discussion.
13) I think it is important to consider the possible age variable in the sample and add it, as well as possible interpretations with studies in other cultural contexts.
14) Limitations are adequate.
15) I miss any additional theoretical or practical implications and a clear conclusion at the end of the discussion.
I believe that the manuscript can be published, although some improvements are needed. I am confident that the authors will improve as they are a very solid research team.
Best regards,
Author Response
INTRODUCTION
It is well written and contains an adequate presentation of the topic of study. The main problem is in the hypothesis approach. I think it would be more appropriate to make several hypotheses (since there are several objectives) and also indicate whether these hypotheses are related to other similar studies (add intratext quotations, if so).
Authors’ answer: thank you for your suggestion. We have now included a hypothesis for each objective and a rationale for each hypothesis based in previous studies (page 3, lines 188-247).
I am also concerned, in relation to Gender Role Theory, about the relationship that may exist with the European-Spanish context. I appreciate that several of the studies are carried out in the USA and gender theory can be closely associated with social/cultural changes. This could be further explored in discussion as well.
Authors’ answer: We have discussed the similarities between USA and Spain regarding gender roles (page 2-3, lines 93-153) and in the discussion as well (page 7-8, lines 460-485).
METHODS
1. Could you specify on line 133 whether it is 230,000 students? The normal thing is to indicate the number of students of the degrees where the sample has been obtained. I do not understand the data.
Authors’ answer: thank you for your comment. Than was a typo. We wanted to indicate that the university population was approximately 23000 undergraduate students. We have now explained more carefully the participant’s selection process considering the total number of male undergraduate students (page 6, lines 250-256)
2. It is important to indicate the type of sampling (I imagine not probabilistic of incidental type).
Authors’ answer: we are sorry about that. You were right, our sampling method was not probabilistic but we calculate the amount of students that should participate in order to have a representative sample of our university.
3. I am concerned about age range. If there is a large sample older than 25-30 years it may be necessary to consider specific analyses, as gender theory may be affected by this variable.
Authors’ answer: we agree with you but we decided not to include additional analysis considering that only 39 participants were older than 25 years. It will be interesting to have a bigger sample in the future and compare the relationships between gender norms and aggressive behavior in different ages.
4. I suggest additional reliability analyses for the sample, as there are some especially low ones (such as the CMNI). They can perform ordinal alpha or omega coefficients. Perhaps all three would give a more complete information.
Authors’ answer: thank you for your suggestion. We have included the McDonald’s Omega in the reliability result of the two measures employed.
5. The indicated time (40 minutes) does not make sense for the indicated questionnaires. It may be a misprint of a more ambitious project with more evaluation tools, but not for 3 questionnaires. Please review it.
Author’s answer: thanks for your comment. It was a mistake. This study was part of a larger project and we indicated the whole time employed to fill other instruments. The average time was 15 minutes for the three scales.
6. I think it's important to further clarify the information on data analysis. I suggest making this point according to each objective and including all the analyses carried out.
It is not clear how the prevalence of the problem is calculated.
Author’s answer: Thank you. We have now rewritten the analysis section to include the procedure followed to calculate the prevalence and indicate the objective of each analysis carried.
It is also not clear (and it appears in more parts) the issue of "bidirectionality". For example, the abstract reads: "bidirectionality of cyber dating abuse by analyzing the extent to which victimization and perpetration co-occur". I believe that using two profiles together is not the same as establishing bidirectionality relationships (which require other analyses).
Author’s answer: you are right. Thank you for your recommendation. Given the cross-sectional nature of the study we have to avoid causality and we opted to use the term co-occurrence, and overlap to avoid reiteration of the terms.
7.On line 176, you indicate "feminine gender". This baffles me very much.
Authors’ answer: thank you for this. It was a typo. We have now modify that sentence.
Results
8 ) I don't understand "Hetero-LGB" in table 1.
Author’s answer: thank you for your comment. We agree that it could be confusing so we have decided to delete it. We tried to indicate that we had two categories regarding sexual orientation: 1) heterosexual and 2) LGB community.
9) I did not find how the prevalence was estimated. Scores equal or above 2? Is that the usual criterion for CDA or is it provided by the original authors?
Authors’ answer: we have now included the procedure followed to calculate prevalence. You can find the description in the analysis section (page 5)
10) Why do you not use control perpetration for the victimization prediction and the same goes for direct victimization and perpetration.
Authors’ answer: Thank you for your comment. We did but it was wrong expressed in the text. We have now revised all the description of results to solve various typos regarding the wording of the study variables.
11) Include the results of the cohen d.
Authors´ answer: we have included cohen’s d.
Discussion
12) If the authors add more hypotheses they should make those adjustments in the discussion.
13) I think it is important to consider the possible age variable in the sample and add it, as well as possible interpretations with studies in other cultural contexts.
15) I miss any additional theoretical or practical implications and a clear conclusion at the end of the discussion.
Authors’ answer: discussion has been modify following your suggestions. Thank you very much for your comprehensive review.
Reviewer 2 Report
I have enjoyed reading this paper. It is well organized. I noticed two things that need to be fixed before publication: on page 4, line 176 you state "feminine gender roles" and I think that is a typo. You study masculine.
Another issue on page 4, line 164, why is this sentence relevant "6 months after the academic year began".
Author Response
1) I have enjoyed reading this paper. It is well organized. I noticed
two things that need to be fixed before publication: on page 4, line 176
you state "feminine gender roles" and I think that is a typo. You study
masculine.
Authors' answer: thank you for you kind and comprehensive review. We have modify the typo.
2) Another issue on page 4, line 164, why is this sentence relevant "6 months after the academic year began".
Authors' answer: We only wanted to indicate the precise time when the recollection of data was made. But it is not really revelant so we have deleted it.
Thank you very much for your review.
Reviewer 3 Report
The present manuscript includes interesting results about cyber dating abuse in a sample of Spanish male university students, exploring the relationships between cyber dating abuse and the internalization in men of masculine gender norms. Currently, new technologies are increasingly common as a form of communication and also as a means to perpetrate aggression (cyberbullying, cyber dating abuse). In this context, this study increases our knowledge about online aggressions in intimate partner relationships.
It is advised to acceptance the publication of this manuscript, with some minor changes.
Introduction
Authors provide sufficient background about cyber dating abuse. Introduction is well organized and the different subsections are adequate. The aim of the research is clearly established by the authors.
Methods
The research design is appropriate. The authors provide sufficient information about the sample and the instruments used to measure the variables included in this study. However, as a suggestion, it would be convenient to indicate some examples of the items included in the Masculine Gender Norms scale. Also, it would be convenient to indicate to what extent masculinity norms are similar in US and Spain. Are these norms similar in both cultural contexts?
Line 176: “feminine gender norms” or “masculine gender norms”?
Results
Results are clearly described, and tables are adequately presented. Only there is a minor question in Table 2.
In Table 2, I think the first number in each column is “n”. So, at the top of each column authors should change “% (n)” to “n (%)”.
Also, it should be explained (possibly in previously section: Analysis data/plan) how authors have established the different groups of perpetrators (perpetration only), victims (victimization only), and perpetrators-victims. What criteria have been followed by authors to assign the participants in this study to each group? Were all items on the scale considered or was a single item sufficient to assign them to a particular group? Was taken into account the frequency indicated by the participants in the different items of the cyber dating abuse scale (never; not in the last year, but before; once or twice; 3 to 10 times; 10 to 20 times; more than 20 times)?
Line 207: “After accounting for control victimization” or “After accounting for direct victimization”?
Discussion
The conclusions are supported by the results, and authors provide interesting data about cyber dating abuse. Nevertheless, I think that it would be interesting to differentiate prevalence data of cyber control and cyber aggression. Co-occurrence in cyber dating abuse is observed (Table 2) in cyber-control behaviors (control abuse), but not in cyber-aggression behaviors (direct abuse). It would be convenient to point out these differences and provide possible explanations for these results.
Also, the results that indicated that being a victim of cyber dating abuse increased the probabilities of being a perpetrator, and vice versa, could be more commentated by authors. Although it is a cross-sectional study, it would be interesting to develop some possible explanations to the correlation observed in this study between cyber dating abuse victimization and perpetration.
Author Response
1) The research design is appropriate. The authors provide sufficient information about the sample and the instruments used to measure the variables included in this study. However, as a suggestion, it would be convenient to indicate some examples of the items included in the Masculine Gender Norms scale. Also, it would be convenient to indicate to what extent masculinity norms are similar in US and Spain. Are these norms similar in both cultural contexts?
Authors’ answer: Thank you for your suggestion. We have discussed the similarities between USA and Spain regarding gender roles (page 2-3, lines 93-153) and in the discussion as well (page 7-8, lines 460-485).
2) Line 176: “feminine gender norms” or “masculine gender norms”?
Authors’ answer: thank you for this. It was a typo. We have now modify that sentence.
3) Results are clearly described, and tables are adequately presented. Only there is a minor question in Table 2.In Table 2, I think the first number in each column is “n”. So, at the top of each column authors should change “% (n)” to “n (%)”.
Authors’ answer: you were right, we have modify the table.
4) Also, it should be explained (possibly in previously section: Analysis data/plan) how authors have established the different groups of perpetrators (perpetration only), victims (victimization only), and perpetrators-victims. What criteria have been followed by authors to assign the participants in this study to each group? Were all items on the scale considered or was a single item sufficient to assign them to a particular group? Was taken into account the frequency indicated by the participants in the different items of the cyber dating abuse scale (never; not in the last year, but before; once or twice; 3 to 10 times; 10 to 20 times; more than 20 times)?
Authors’ answer: we apologize for this. We missed the explanation. we have now included the procedure followed to calculate prevalence following the procedure used by the authors of the scale. You can find the description in the analysis section (page 5)
5) Line 207: “After accounting for control victimization” or “After accounting for direct victimization”?
Authors’ answer: Thank you for your comment. We have now revised all the description of results to solve various typos regarding the wording of the study variables.
6) The conclusions are supported by the results, and authors provide interesting data about cyber dating abuse. Nevertheless, I think that it would be interesting to differentiate prevalence data of cyber control and cyber aggression. Co-occurrence in cyber dating abuse is observed (Table 2) in cyber-control behaviors (control abuse), but not in cyber-aggression behaviors (direct abuse). It would be convenient to point out these differences and provide possible explanations for these results.
Authors’ answer: Thank you for your comment. We found co-occurrence in both forms of abuse. The text could be mistaken because we only included a table for those forms where (although small) significant effects were found about the relationships of feminine gender norms and cyber dating abuse. However, victimization accounted for a high percentage of perpetration (and vice versa) in both types of abuse. We have added that explanation in the text.
Following your advice, we have included a sentence in the discussion to point the higher prevalence of control forms of abuse in comparison with direct forms. (page 7, lines 397-400)
7) Also, the results that indicated that being a victim of cyber dating abuse increased the probabilities of being a perpetrator, and vice versa, could be more commentated by authors. Although it is a cross-sectional study, it would be interesting to develop some possible explanations to the correlation observed in this study between cyber dating abuse victimization and perpetration.
Author’s answer: you are right. Thank you for your recommendation. Given the cross-sectional nature of the study we have to avoid causality and we opted to use the term co-occurrence, and overlap to avoid reiteration of the terms. We have also discussed more carefully (page 6, lines 404-410).
Thank you very much for your review. We appreciate your careful reading.
Round 2
Reviewer 1 Report
Dear Authors
Thank you very much for your review.
I think the manuscript has improved.
I'm glad I helped in the process.
Congratulations on your study.
I will recommend your acceptance.
Best regards,
Reviewer 3 Report
All questions have been adequately answered.
It is an interesting study, and I advise its publication.
Best regards,